# Comparison of Long Short Term Memory Networks and the Hydrological Model in Runoff Simulation

**Hongxiang Fan** [1,2] , **Mingliang Jiang** [1,2], **Ligang Xu** [1,2,3,*] , **Hua Zhu** [1,2] , **Junxiang Cheng** [1,2] and **Jiahu Jiang** [1,2]

1   Key Laboratory of Watershed Geographic Sciences, Nanjing Institute of Geography and Limnology, Chinese Academy of Sciences, Nanjing 21008, China; hxfan@niglas.ac.cn (H.F.); jiangmingliang18@mails.ucas.ac.cn (M.J.); zhuhua18@mails.ucas.ac.cn (H.Z.); chengjunxiang15@163.com (J.C.); jiangjh@niglas.ac.cn (J.J.)
2   University of Chinese Academy of Sciences, Beijing 100049, China
3   China Three Gorges Corporation, Eco-Environmental Engineering Research Center, Beijing 100038, China
*   Correspondence: lgxu@niglas.ac.cn; Tel.: +86-025-8688-2105

**Abstract:** Runoff modeling is one of the key challenges in the field of hydrology. Various approaches exist, ranging from physically based over conceptual to fully data driven models. In this paper, we propose a data driven approach using the state-of-the-art Long-Short-Term-Memory (LSTM) network. The proposed model was applied in the Poyang Lake Basin (PYLB) and its performance was compared with an Artificial Neural Network (ANN) and the Soil & Water Assessment Tool (SWAT). We first tested the impacts of the number of previous time step (window size) in simulation accuracy. Results showed that a window in improper large size will dramatically deteriorate the model performance. In terms of PYLB, a window size of 15 days might be appropriate for both accuracy and computational efficiency. We then trained the model with 2 different input datasets, namely, dataset with precipitation only and dataset with all available meteorological variables. Results demonstrate that although LSTM with precipitation data as the only input can achieve desirable results (where the NSE ranged from 0.60 to 0.92 for the test period), the performance can be improved simply by feeding the model with more meteorological variables (where NSE ranged from 0.74 to 0.94 for the test period). Moreover, the comparison results with the ANN and the SWAT showed that the ANN can get comparable performance with the SWAT in most cases whereas the performance of LSTM is much better. The results of this study underline the potential of the LSTM for runoff modeling especially for areas where detailed topographical data are not available.

**Keywords:** LSTM; runoff simulation; Poyang Lake Basin; deep learning

## 1. Introduction

Understanding the change in regional hydrological cycles under a changing climate is an important task, not only for regional water resources management but also for the sustainable development of society [1,2]. One of the key challenges in the filed of hydrology is to simulate runoff accurately [3]. Many studies aim to solve this problem with different approaches. These approaches can be classified into three categories—(1) conceptual model, consisting concepts used to simulate runoff [4]; (2) physically based model, representing the real runoff generation process and (3) data-driven method, estimating runoff from the bunch of input variables [5].

The physically based model requires major efforts on model calibration and can lead to remarkably different results because of the uncertainty in model structure and parameter estimation [6–9]. Moreover, accurate simulation for hydrological or hydrodynamic process is highly computational

intensive and therefore limited to simulations of restricted duration [10]. Consequently, simple and practical user-friendly methods for quick runoff simulations and predictions with minimum data requirement is needed.

Previous studies demonstrated the feasibilities of the data-driven methods, like Multiple Linear Regression (MLR) [11], Support Vector Machines (SVM) [12] and Artificial Neural Networks (ANN) [10], in hydrological applications and the comparisons between traditional physical models were also discussed. It is noticed that the machine learning methods can achieve comparable or even better performances compared to traditional physically based models in many cases, including water level prediction [10], groundwater simulation [13], soil moisture retrieving [14] and runoff modeling [15]. However, traditional machine learning methods like ANNs are not designed specially for handling time series data, which can not maintain time dependencies directly [16]. Recently, the advantages of deep learning (DL) over traditional machine learning (ML) methods have been repeatedly proved by scientific communities, transforming not only daily lives but also various scientific disciplines [17]. As defined by Chollet and Allaire [18], "DL is a specific subfield of machine learning: a new take on learning representations from data that puts an emphasis on learning successive layers of increasingly meaningful representations." DL can beat traditional ML methods due to its abilities to (1) extract and shape a cascade of abstract features from raw data; (2) transfer learning after the knowledge is learned; (3) represent complex functions with increased network depth [17]. Two state-of-the-art, widely adopted DL architectures are convolutional neural networks (CNNs) for images tasks and long short-term memory (LSTM) for time series simulation. The applications of DL can be found in fields like computer vision [19,20], speech recognition [21] and natural language processing [22].

However, the adoption of DL in water resources or hydrology has so far only been gradual. To our knowledge, Fang et al. [16] were the first to exploit LSTM's ability to build dynamical hydrological models with forcings and memory, indicating that LSTM generalizes well across regions with distinct climates and environmental settings in predicting soil moisture. Another study conducted by Kratzert et al. [3] trained LSTM models to predicts runoffs for hundreds of basins over the U.S. without providing differentiating physical factors. Their results suggest that single basin data was not sufficient to train the model. In addition, Hu et al. [23] compared the capability of LSTM and ANN in rainfall-runoff simulation, indicating the special units of forget gate in LSTM is more suitable for flood forecasting. All aforementioned studies have presented the potential of LSTM in hydrological modeling applications. Technically, time series simulation or forecasting can be phrased as a supervised learning problem. Supervised learning is to learn the mapping function from the input ($X$) to the output ($Y$) variables. Given a time series dataset, we can use previous time steps as input variables and the next time step as the output variable. The number of previous time steps is called the window size. Consequently, the window size may have impacts on the forecast accuracy and should be selected carefully. However, previous studies like those by Kratzert et al. [3] and Zhang et al. [24] failed to analyze this kind of impacts by setting window size as a somewhat arbitrary number. The impacts of window size on the runoff simulation performance need to be further analyzed. In addition, previous studies usually take the precipitation as the only input, neglecting the effects of other meteorological variables on runoff generation. Although the simulation of runoff based on LSTM with only precipitation data can achieve desirable results in some cases, whether it can be further improved with more meteorological variables as input is still vague. Moreover, as neural network like LSTM is data-driven methods, its capability compared to traditional hydrological model is yet to be analyzed.

To this end, this study aim to build a simple LSTM neural network, training it with both precipitation data and other meteorological data. Its performance is then compared with traditional hydrological and ANN models. We choose Poyang Lake Basin (PYLB) as our study area due to its heterogeneous hydrological characteristics across space, providing a good example for us to test our model performance. More details of PYLB can be found in Section 2.1. The objectives of this study are to: (1) analyze the effects of window size on the performance of simulation; (2) analyze whether the

simulation performance can be improved by inputing more meteorological variables; (3) compare the simulation capability of LSTM model with other models (i.e., traditional ANN and SWAT models). The results of this study can provide better understandings of the potential of LSTM in hydrological simulation applications.

## 2. Study Area and Data Source

### 2.1. Study Area

Poyang Lake is the largest freshwater lake in China, which is located on the middle of the Yangtze River [25]. The drainage basin area of Poyang Lake is 162,225 km$^2$ (Figure 1), covering about 97% of Jiangxi province [26]. Poyang Lake exchanges water with the Yangtze River through a narrow outlet located in Hukou [27], while receiving water from the five sub-basins of the Ganjiang River, the Fuhe River, the Xinjiang River, the Raohe River and the Xiushui River [25]. The PYLB belongs to a subtropical wet climate zone with an annual mean precipitation of 1680 mm and annual mean temperature of 17.5 °C [28]. In normal years, the water surface of Poyang Lake can expand to 4000 km$^2$ and the water volume is about $3.20 \times 10^{10}$ m$^3$. While in dry season, the lake will shrink to little more than a river. The dramatic seasonal water-level fluctuations of Poyang Lake shape the unique landscape that has been described as "flooding like the sea, drying like thread" [25,29].

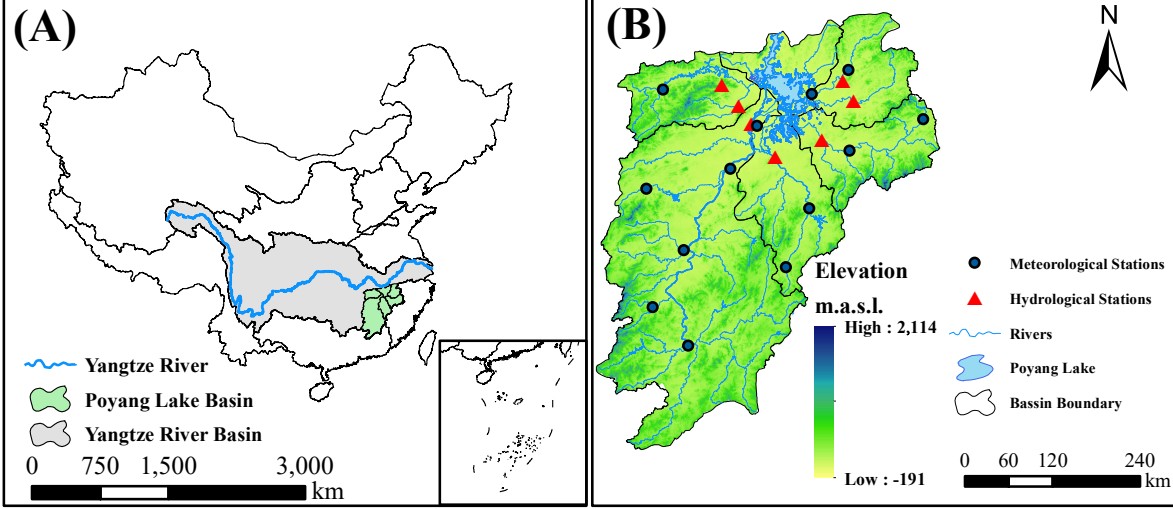

**Figure 1.** Topography and river networks of Poyang Lake Basin (**A**). Location of 7 hydrological stations and 13 meteorological stations are presented with red triangles and blue dots, respectively (**B**).

### 2.2. Data Source

Daily runoff of seven hydrological stations inside the PYLB were obtained from Management Bureau of the Yangtze River (MBYR) catchment for period from 1960 to 2013. The sub-basins' runoff was represented by the average of the gauged runoff inside each sub-basin. The basic features of these hydrological stations are listed in Table 1.

The meteorological data from 13 National Meteorological Observatory (NMO) stations inside the PYLB from 1960–2013 were obtained from the National Climate Center of China Meteorological Administration (CMA), including daily observations of precipitation, temperature, relative humidity and so on. Missing values in the meteorological data were further interpolated using data from the nearest 5 meteorological stations [28].

**Table 1.** List of hydrological stations used in this study.

| Hydrological Station | Location | Coordinates | Gauged Area (km$^2$) |
|---|---|---|---|
| Qiujin | Xiushui | 115.41° E , 29.10° N | 9914 |
| Wanjiabu | Xiushui (Liaohe tributary) | 115.65° E, 28.85° N | 3548 |
| Waizhou | Ganjiang | 115.83° E, 28.63° N | 80,948 |
| Lijiadu | Fuhe | 116.17° E, 28.22° N | 15,811 |
| Meigang | Xinjiang | 116.82° E, 28.43° N | 15,535 |
| Hushan | Raohe (Le'an tributary) | 117.27° E, 28.92° N | 6374 |
| Dufengken | Raohe (Changjiang tributary) | 117.12° E, 29.16° N | 5013 |

*2.3. Data Pre-Processing*

Standardization of datasets is a common requirement for many neural network frameworks [18]. Consequently, a zero-mean normalization was applied to all the climatic and hydrological data, ensuring the data remain on the same scale and further guaranteeing a quick and stable convergence of the parameters inside the model. The formula for zero-mean normalization is as follows:

$$\hat{x_{ij}} = \frac{x_{ij} - \bar{x}_j}{\sigma_j}, \tag{1}$$

where $x_{ij}$ denotes the $j$th original data in time $i$ and $\hat{x_{ij}}$ represents the normalized data for $j$th variable in time $i$. $\bar{x}_j$ and $\sigma_j$ denote the average and standard deviation for $j$th variable in time $i$.

Unlike the traditional hydrological models, the samples in the neural network are not necessarily ordered chronologically [3]. Previous research demonstrated that it is helpful to have random samples for faster convergence [30]. Consequently, we shuffle the data randomly before feeding to the neural network.

**3. Methodology**

*3.1. Long Short Term Memory Network*

Long Short Term Memory (LSTM) network is special kind of recurrent neural network (RNN) structure, overcoming the weakness of the traditional RNN to learn long-term dependencies. It was first introduced by Hochreiter and Schmidhuber [31] and was refined and popularized by a lot of researchers [32]. The"deep in time" structure of LSTM enables it to lean when to forget and how long to retain the state information through the specially designed *gates* and *memory cells* (Figure 2).

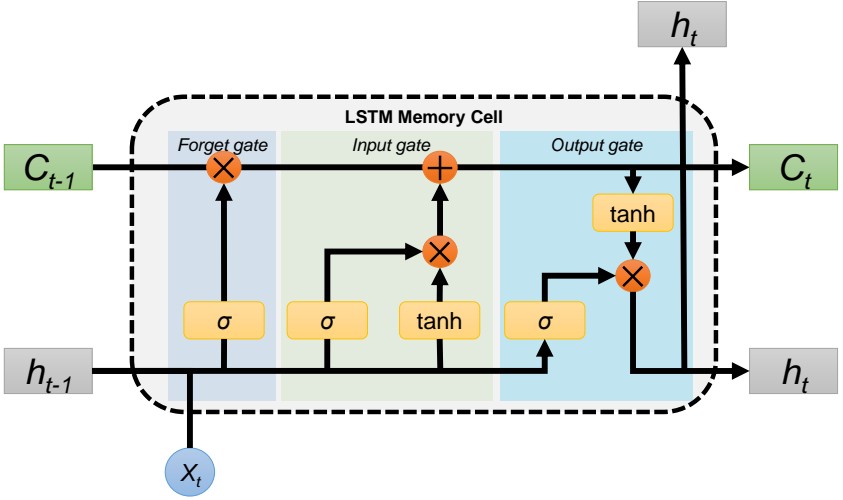

**Figure 2.** The architecture of Long Short Term Memory (LSTM) cell.

The key to LSTM is the cell state ($C_t$), which enables the information to flow along it unchanged. The cell state is regulated by three gates to optionally let information through. The first gate is called the forget gate, controlling which elements of the cell state vector $C_{t-1}$ will be forgotten.

$$f_t = \sigma(W_f \cdot [h_{t-1}, x_t] + b_f), \tag{2}$$

where $f_t$ is an output vector of the sigmoid layer with values ranging from 0 to 1, indicating the forgotten degree. $W_f$ and $b_f$ define the set of trainable parameters for the forget gate.

Following that, the input gate decides which value to be updated:

$$i_t = \sigma(W_i \cdot [h_{t-1}, x_t] + b_i), \tag{3}$$

where $i_t$ is an output variable with value ranging from 0 to 1. $W_i$ and $b_i$ are trainable parameters.

Then a potential vector of cell state is computed by the current input ($x_t$) and the last hidden state $h_{t-1}$:

$$\tilde{C} = \tanh(W_C \cdot [h_{t-1}, x_t] + b_C), \tag{4}$$

where $\tilde{C}$ is a vector with values ranging from 0 to 1, tanh is the hyperbolic tangent and $W_C$ and $b_C$ are the trainable parameters.

After that, we can update the old cell state $C_{t-1}$ into the new cell state $C_t$ by element-wise multiplication:

$$C_t = f_t * C_{t-1} + i_t * \tilde{C}_t. \tag{5}$$

Finally, the output gate decides which to be output by a sigmoid layer:

$$o_t = \sigma(W_o[h_{t-1}, x_t] + b_o), \tag{6}$$

where $o_t$ is a vector with values ranging from 0 to 1. $W_o$ and $b_o$ are trainable parameters defined for the output gate.

The new hidden state $h_t$ is then calculated by combining Equations (5) and (6):

$$h_t = o_t * \tanh(C_t). \tag{7}$$

### 3.2. LSTM Setup

In this work, we developed a basic neural network model based on the LSTM framework. The model contains 3 layers (Figure 3). One LSTM layer with 128 LSTM neurons was set as the input layer. A dropout layer was set on the first LSTM layer with a dropout rate of 0.4. A fully connected layer was then set up as the output layer which yields 5 distinct runoff time series. Two input dataset (i.e., $D_1$ and $D_2$) were used to train the model. $D_1$ contains only the precipitation data, whereas $D_2$ contains all available meteorological data (e.g., precipitation, air temperature, relative humidity and so on). The model trained using $D_1$ is then called LSTM$_1$ whereas the other one which was trained with $D_2$ is named as LSTM$_2$. For each LSTM model, the train period is from 1 January 2002 to 31 December 2008 and the test period is from 1 January 2009 to 31 December 2013.

### 3.3. ANN Setup

The ANN implemented in this study was a standard three-layer feed-forward networks [10]. It contains a hyperbolic tangent sigmoid transfer function in the hidden layer and a linear transfer function in the output layer. The model structure is shown in Figure 4. Similar as the proposed LSTM model, the ANN model also contains 3 layers. One fully connected layer with 128 neurons was set as the input layer. A dropout layer was set on the first layer with a dropout rate of 0.4. A fully connected layer was then set up as the output layer which yields 5 distinct runoff time series. The model was

trained using $D_2$ and the train period is from 1 January 2002 to 31 December 2008 and the test period is from 1 January 2009 to 31 December 2013.

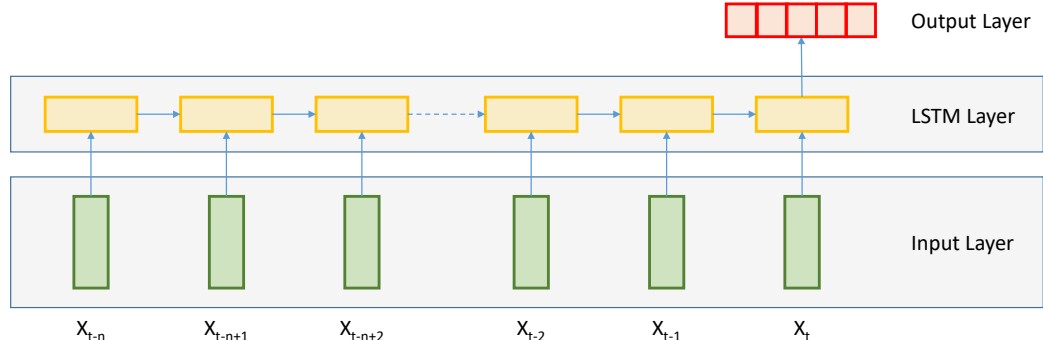

**Figure 3.** The architecture of the proposed LSTM model.

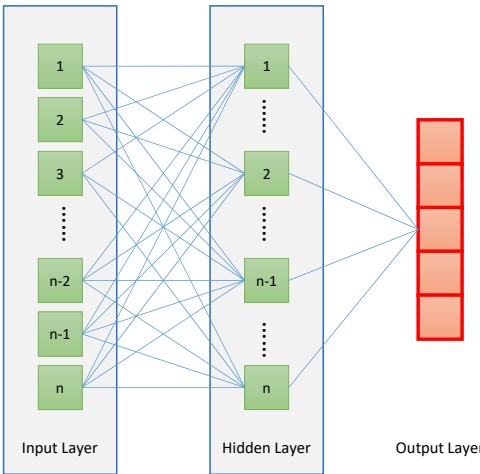

**Figure 4.** The architecture of the proposed Artificial Neural Network (ANN) model.

### 3.4. Tuning Procedure for LSTM

#### 3.4.1. Tuning for Window Size

As mentioned before, the window size should have impacts on the model performance thus it is important to be analyzed. In this study we tried to illustrate the impacts of window size from a very short period to a relative long period (i.e., from a couple of days to half year). However, due to the limitation of both time and computational capability, we compromised by presenting results for the selected eleven window sizes, which are 1, 5, 10, 15 20, 25, 30, 60, 90 and 180 days. The daily and weekly impacts of meteorological variables on runoff generation can be illustrated by window sizes like 1 to 25 days, whereas the monthly impacts can be illustrated by window size 30 days. The other window sizes like 60, 90 and 180 days were added to see if the daily runoff process is affected by inter-annual climate variability. The window size which yields the best model performance is then selected for further application.

#### 3.4.2. Tuning for Hyperparameters

The neural networks often contain lots of hyperparameters, whose value is set before the learning process begins. Hyperparameter optimization or tuning is the process to find a tuple of hyperparameters that yields a model which minimize the loss function on the given data [19]. In this

study, we used the mean-square-error as the loss function for hyperparameter optimization according to Kratzert et al. [3].

Common hyperparameters include learning rate, training epochs, the dimensionality of the output space and so on [19]. Learning rate is a hyperparameter which represents the step size in a gradient descent method [33,34]. In this study, we use the efficient Adam version of stochastic gradient descent [35]. The initial learning rate was set to be 0.2 and a time-based decay rate was used to update the learning rate through the training process. In addition, the epoch, which is generally defined as one pass over the entire dataset in neural network, is used to separate training into distinct phases. Training for too long will lead the model to be overfit, learning patterns that only exist in the training dataset. Whereas training for too short will lead the model to be underfit, which means the model has not learned the relevant patterns in the training data [19]. In this study, the training epochs were first set to 200 according to Kratzert et al. [3] and the highest NSE were achieved after 50 epochs. Consequently, we use the number of 50 epochs for the final training of the model. Other hyperparameter like the the dimensionality of the output space is tuned using grid search method in our study [18,19].

*3.5. Evaluation Metrics*

In this study, the metrics for evaluating model performance are root mean square error (RMSE) and the Nash-Sutcliffe Efficiency (NSE). The calculation procedure of the 2 metrics are as follows:

$$RMSE = \sqrt{\frac{\sum_{i=1}^{n}(y_i - y_i')^2}{n}} \tag{8}$$

$$NSE = 1 - \frac{\sum_{i=1}^{n}(y_i - y_i')^2}{\sum_{i=1}^{n}(y_i - \bar{y})^2}, \tag{9}$$

where $y_i$ and $y_i'$ denote the observed and simulated runoff at time $i$. $\bar{y}$ and $\bar{y}'$ denote the average observed and simulated runoff at time $i$.

*3.6. Hydrological Model*

To compare the performance of LSTM with traditional distributed hydrological model, the Soil & Water Assessment Tool (SWAT) was used to set up the model to simulate runoff of Poyang Lake Catchment. SWAT is a widely used distributed hydrological model [36,37], which is implemented with SUFI-2 algorithm to optimize model's parameters [38–40]. SWAT provides a simple framework for runoff simulation by discretizing the landscape into hydrological response units (HRUs). It first simulates the hydrological process at the HRU level and subsequently routed at the sub-basin level. Water balance equations are applied at the HRU level [41]. The model was fed with meteorological data from 14 National Meteorological Observatory (NMO) stations inside the PYLB were obtained from the National Climate Centre of China Meteorological Administration (CMA), including daily observations of air temperature, wind speed, relative humidity, sunshine hours and absolute vapor pressure, among others. The land use data obtained from the Department of Soil Survey of Jiangxi Province was categorized into 6 types, namely, farmland (26.72%), forest (61.82%), pasture (4.33%), water body (4.19%), urban (2.54%) and others (0.40%). The soil data inside the basin was classified according to the Genetic Soil Classification of China, and the main 6 soil types are red soil (55.25%), paddy soil (23.06%), yellow soil (15.40%), purple soil (1.58%), alluvial soil (0.90%) and calcareous soil (0.73%). The basin was delineated using a 90-m digital elevation model (DEM). Delineation and initial model parameterization were achieved using the ArcSWAT interface with the input datasets previously described. Based on previous experience in modeling this watershed, 1000 ha were used to define the minimum area necessary for channel formation. The HRU threshold for soils and land use was set to 0%. Watershed delineation produced 38 subbasins consisting of 754 HRUs (Figure 5). The model ran at daily time step from 2000–2013. The first two years served as a warm-up period to initial condition for model experiments. The simulated runoff was calibrated against the daily

observed runoff in the period of 2002–2008 and validated in the period of 2009–2013. The calibration and validation procedure and other details of SWAT model for PYLB can be found in Guo et al. [42].

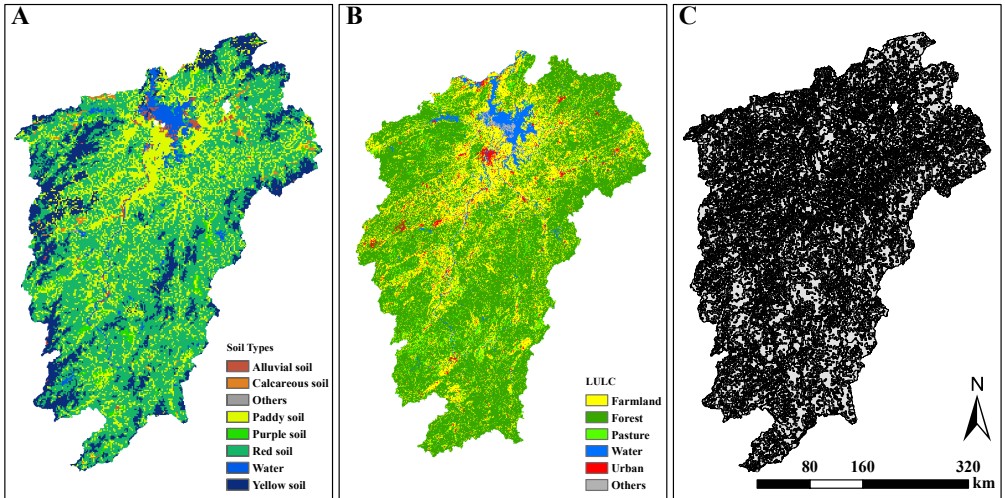

**Figure 5.** Characteristics of the Poyang Lake Basin: (**A**) Soil types; (**B**) Land use and (**C**) the hydrological response units.

## 4. Results

### 4.1. Effect of the Window Size

Here we analyzed the potential effects of window size on the model performance (which is represented by RMSE and NSE). As mentioned above, only the results for the selected window sizes (i.e., 1, 5, 10, 15 20, 25, 30, 60, 90, 180 days) are presented. The comparison results are shown in Figure 6. As shown in Figure 6, RMSE of the LSTM model decreased quickly when the window size increased from 1 to 15 days. Following that, the RMSE started to increase till the window size is up to 60 days. When the window size is lager than 60 days, the RMSE kept decreasing as window size kept being longer. Similar patterns can be found in NSE, that the NSE kept increasing when the window size increased from 1 to 15 days, followed by a decline when the window size is less than 60 days. When the window size is larger than 60 days, although there exist slight improvements in performance in some cases, the overall performance is worse than that when window size is around 15 days. It can be found in Figure 6 that the lowest RMSE and highest NSE are both obtained at window size 15, indicating that a 15-day window size may be the best choice for simulating runoff in PYLB in terms of accuracy and also the computational efficiency.

### 4.2. Overall Performance of LSTMs

The statistical results for the overall performances of LSTM models for both training (calibration) and test (validation) period are listed in Table 2. As shown in Table 2, LSTMs both performed well during the training period, with average NSEs of 0.86 and 0.90 for $LSTM_1$ and $LSTM_2$, respectively. Although $LSTM_1$ is able to follow closely to $LSTM_2$ during the training period, the differences of performances between the 2 models are notable during the test period, especially in Xiushui sub-basin. The NSE for $LSTM_2$ in Xiushui sub-basin was 0.74 during the test period compared to that of $LSTM_1$, which was only 0.60, indicating that meteorological data can improve the model performance.

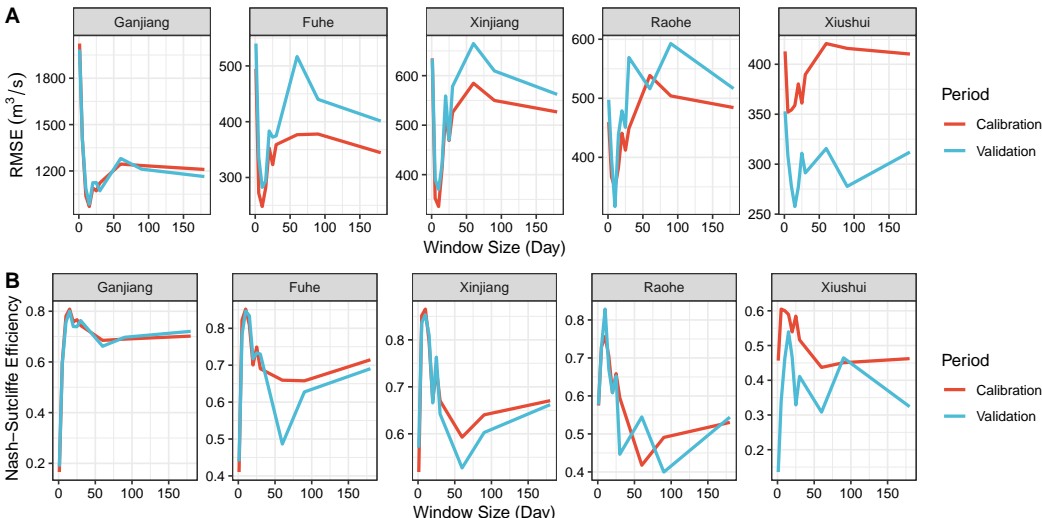

**Figure 6.** Model performance with different window size ((**A**) root mean square error; (**B**) Nash-Sutcliffe Efficiency).

**Table 2.** Performances of LSTMs in runoff simulation.

| Model | LSTM1 | | | | LSTM2 | | | |
|---|---|---|---|---|---|---|---|---|
| Period | Calibration Period | | Validation Period | | Calibration Period | | Validation Period | |
| Metrics | NSE | RMSE | NSE | RMSE | NSE | RMSE | NSE | RMSE |
| Ganjiang | 0.86 | 834.63 | 0.84 | 888.09 | 0.91 | 667.71 | 0.89 | 729.95 |
| Fuhe | 0.91 | 192.26 | 0.91 | 221.60 | 0.92 | 177.87 | 0.91 | 213.89 |
| Xinjiang | 0.92 | 258.20 | 0.92 | 276.36 | 0.94 | 230.78 | 0.94 | 240.18 |
| Raohe | 0.86 | 264.37 | 0.87 | 278.77 | 0.91 | 212.95 | 0.93 | 204.10 |
| Xiushui | 0.73 | 293.72 | 0.60 | 240.41 | 0.83 | 230.79 | 0.74 | 192.16 |

Figures 7 and 8 show the measured and simulated runoff from the LSTM models. For illustrative purpose, only the results of validation period are presented. As shown in Figures 7 and 8, both LSTM$_1$ and LSTM$_2$ can capture the temporal pattern of runoff well for the entire validation period. Most of the peaks have been captured by the LSTMs in sub-basins except for Xiushui, although there still exist some under- or over-estimates. While LSTMs are also behavioral in Xiushui sub-basin, we notice that LSTM$_1$ often noticeably overestimates variations when runoff is relatively low and underestimates the value of peak runoffs. Compared to LSTM$_1$, the performance of LSTM$_2$ in Xiushui sub-basin is much better, with more accurate estimation of peak values.

### 4.3. Comparison of Simulation Capability with Other Models

In terms of simulation capability, time series generated by LSTMs compared favorably against traditional ANN and SWAT across all sub-basins. The performances of different models are summarized in Figure 9. As shown in Figure 9, LSTMs occupied the lowest RMSEs and the highest NSEs during the entire validation period across all sub-basins compared to the traditional ANN and SWAT models. Although the LSTM models seems to have trouble in simulating runoff for Xiushui sub-basin, where LSTM$_1$ scores a NSE lower than 0.60 and the performance can be improved when inputing more meteorological variables, that is, the NSE is 0.74 for LSTM$_2$. In sub-basins except for Xiushui, traditional ANN and SWAT get comparable performance, with ANN being sightly better than the SWAT. However in the Xiushui sub-basin, the performance of the traditional ANN model decreased dramatically, with a NSE lower than 0.40.

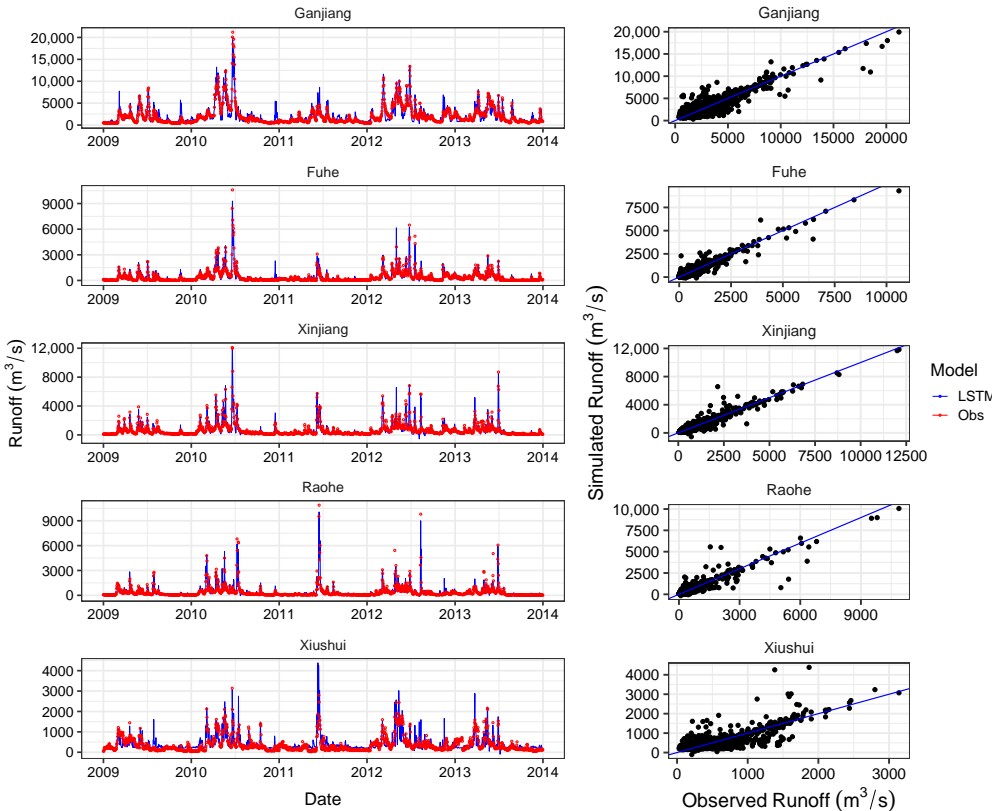

**Figure 7.** Performance of LSTM$_1$ during the validation period for all five sub-basins.

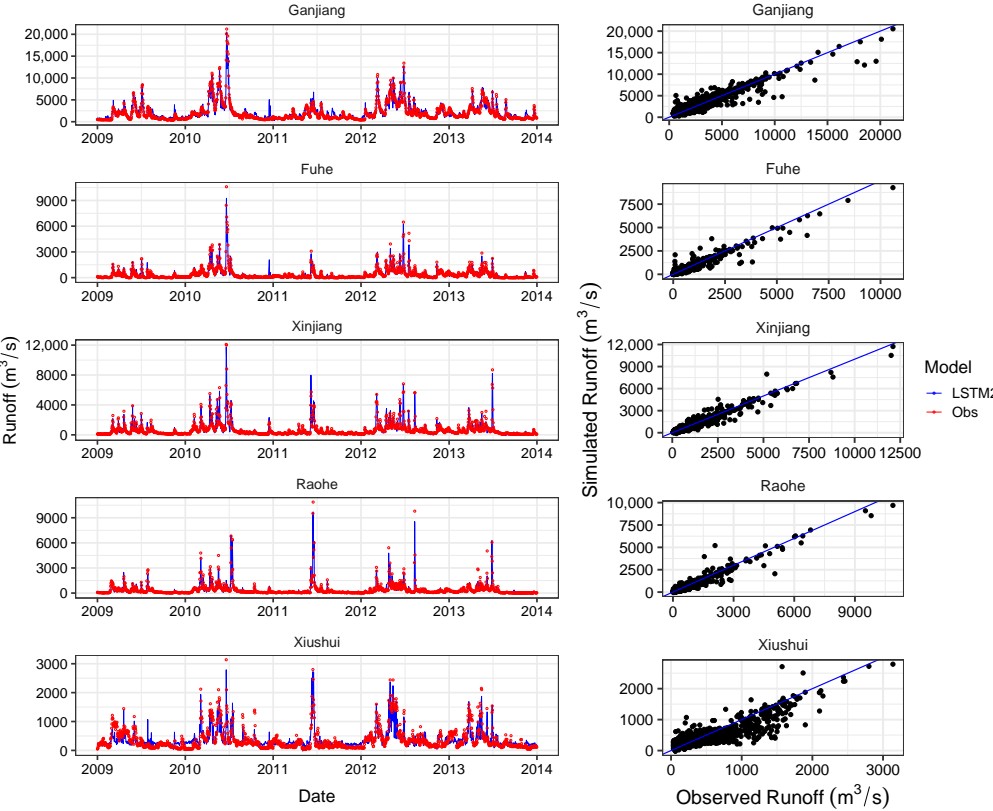

**Figure 8.** Performance of LSTM$_2$ during the validation period for all five sub-basins.

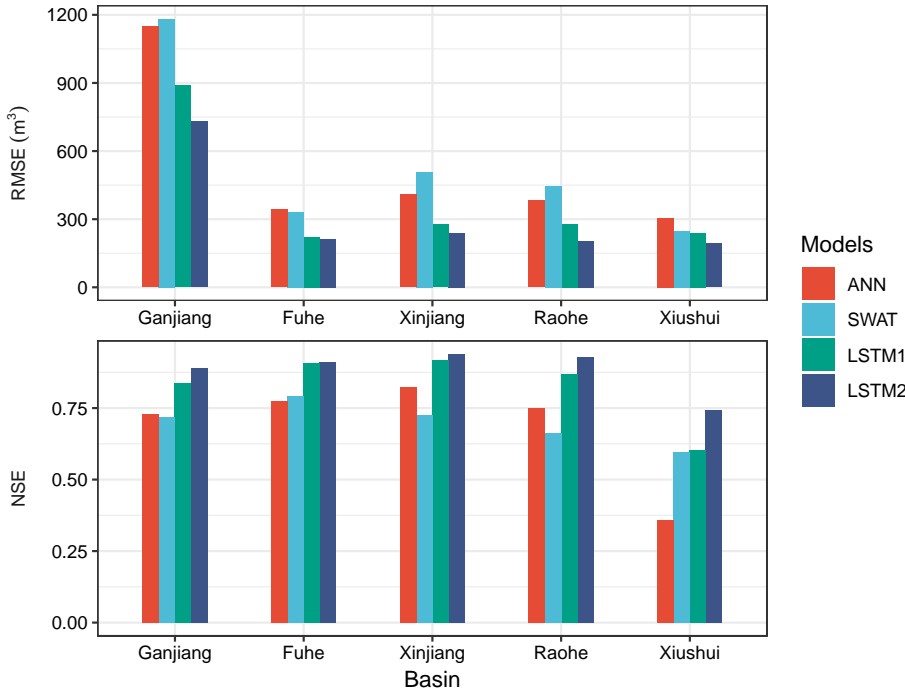

**Figure 9.** Performances of LSTMs, ANN and Soil & Water Assessment Tool (SWAT) model in simulating the runoff.

## 5. Discussion

### 5.1. Potential Factors that Influence Runoff Simulation

In this study, the runoff process across PYLB was simulated using meteorological variables with the aid of neural network models, that is, the LSTM and the ANN. Compared to distributed hydrological model, the LSTM model achieved comparable or even better performance in most sub-basins. However, both the LSTM and the ANN model seemed to have some troubles in simulating runoff for Xiushui sub-basin. As neural network model forecasts the time series using the information learned from the input datasets. The quality and quantity of training data strongly affects the model performance [17,19]. It is noted that only one station is inside Xiushui sub-basin (Figure 1). As runoff process is highly impacted by the regional climate conditions [43], where the data from meteorological station inside the sub-basin will occupy larger weights compared to others. Although we trained the model with all available data from 13 meteorological station (256,412 samples in total), the ultimate output of is the sum of input variables multiplied by the corresponding weights. Consequently, the only meteorological station data maybe fail to represent the climate conditions in Xiushui sub-basin, which may lower the capability of the neural network model for simulating the runoff process inside it. Moreover, the long-term dependencies between the provided input and output cannot be learned by the ANN, which also deteriorates its performance compared with the LSTM, especially in regions where the meteorological data is sparse.

Besides, under the background of global warming, increased climate change was regarded as the dominate factor influencing regional hydrological cycles, with anthropic activities played a complementary role [28,29,42]. Due to rapid developments of urbanization and industrialization, PYLB has undergone intensive human activities since 1950s, exerting considerable impacts on catchment hydrology [29]. The biggest reservoir in Jiangxi Province, that is, Zhelin Reservoir, is located in Xiushui sub-basin with a drainage area of 9340 km$^2$ and the total reservoir capacity of 7.92 billion m$^3$ [44]. The reservoir will notably change the regional hydrology cycle inside Xiushui sub-basin, causing the malfunction of neural network in Xiushui.

*5.2. Limitations and Uncertainties of Neural Network*

Although the technology of neural networks is transforming many scientific disciplines, its applications in hydrology have been few. Shen [17] classified these applications into three categories: (1) extracting hydrological information from images; (2) modeling of hydrological process using observed data; (3) learning and generating complex data distributions. However, the main drawback is that the DL models can not be interpreted from a hydrological perspective and thus are often criticized for their "black-box-ness". Recently, Kratzert et al. [3] argued that by comparing the LSTM cell states with meteorological variables as well as the dynamic catchment properties the controlling process might be revealed, the reasonability is yet to be proved. The model architecture is also important for simulation performance. As Chollet and Allaire [18] point out that pick the right network architecture is more an art than science. The only thing you can rely on to choose the best architecture is practice, making the performance of the model unpredictable. We have tried our best to optimize our model and it achieved better performance compared to other models. However, whether the performance can be improved by applying different model structure is yet to know. In addition, once the model is set up, the huge amount of hyperparameters still make the optimization process intractable. In this study, we were only able to analyze the effects of some key hyperparameters on the model performance. Other hyperparameters such as the number of layers, size of the cell memory and the dropout rate were not specifically optimized but were kept constant, after an initial screening. For optimal performance, these settings should be systematically investigated using an independent validation set. A hyperparameter search of this scale was out of the scope of this study and will be an important task for future studies Moreover, the model performance will be affected by the random initial parameters, adding the randomness of the model. New technologies such as GPU computation and CPU parallelism can result in non-determinisitc execution patterns [45]. Although we have set a fix random seed before initializing the model in order to obtain reproducible results, there still exit some differences between distinct training processes. Such kind of randomness may somehow deteriorate the model feasibilities, especially when training with new data. Consequently, the effect of initial state of the model on the final performance should be analyzed in the further studies.

*5.3. Implications for Regional Water Resources Management*

This study investigated the responses of runoff process to meteorological variables across the PYLB using the state-of-the-art deep learning Long-Short-Term-Memory networks. The excellent performance of LSTM compared to traditional distributed hydrological model (SWAT) indicated that the proposed LSTM can be regarded as an efficient tool for runoff simulation. Recently, as many high-level neural networks APIs like Keras [46] or Pytorch (http://pytorch.org) have been developed, the implementation for a sophisticate neural network is becoming more and more convenient, making it available for scientists or decision-makers who are not specialized in computer science. In addition, compared to traditional hydrological model, the fully data-driven LSTM model consumes less computational power and thus much faster in calibration and simulation, making it possible for the decision-making process of critical situations like early-warning for flood or drought events.

**6. Conclusions**

In this study, we propose a data driven approach using the state-of-the-art LSTM network. The proposed model was applied in the PYLB. The impacts of the number of previous time step and the input datasets in simulation accuracy is tested and the model performance was compared with an Artificial Neural Network and the Soil & Water Assessment Tool.

From the results obtained in this study, the following conclusions can be made:

1.　The performance of the proposed LSTM model is strongly affected by the widow size. A window in improper large size will dramatically deteriorate the model performance. In terms of PYLB, better model performance can be achieved by increasing the window size when it is less than

15 days. When the window size is between 15 and 60 days, the model performance will remain stable as the window size increase. A window size of 15 days might be appropriate for both accuracy and computational efficiency.

2. The proposed LSTM model can achieve desirable results (where the NSE ranged from 0.60 to 0.92 for the test period) with precipitation data as the only input. And the performance can be improved simply by feeding the model with more meteorological variables (where NSE ranged from 0.74 to 0.94 for the test period). In addition, the LSTM model with more meteorological variables is able to capture the peak values of runoff more precisely.

3. The comparison results with the ANN and the SWAT showed that the the proposed LSTM model scored the best in most cases even in regions with sparse meteorological stations. The application of LSTMs and its further development have therefore a high potential to extend data-based modeling approaches in the field of hydrology.

**Author Contributions:** Conceptualization, H.F. and L.X.; methodology, H.F.; software, H.F.; validation, H.F., M.J., H.Z. and J.C.; formal analysis, H.F.; investigation, H.F.; resources, H.F.; data curation, H.F.; writing—original draft preparation, H.F.; writing—review and editing, H.F.; visualization, H.F.; supervision, J.J.; project administration, L.X.; funding acquisition, L.X. All authors have read and agreed to the published version of the manuscript.

**Funding:** The research was funded by National Key Research and Development Program of China (2018YFE0206400, 2018YFC0407606), the National Natural Science Foundation of China (41971137 and 41771235), the STS Key Projects of the Chinese Academy of Sciences (KFJ-STS-QYZD-098), the Science and Technology Planning Project of Qinghai Province (2019-HZ-818), China Three Gorges Corporation (01903145), Nanjing Institute of Geography & Limnology (NIGLAS2019QD005), the Sichuan Provincial Science and Technology Department (2018GZ0499) and the Education Department of Sichuan Province (17ZB0399).

**Acknowledgments:** We would like to thank the Editor, Associate Editor and two anonymous reviewers for their constructive comments, which have noticeably improved the final manuscript. We would like to thank Rong Wang for her tremendous help in setting up the SWAT model. The great help of Tao Huang in building the LSTM model is also appreciated.

**Conflicts of Interest:** The authors declare no conflict of interest.

## Abbreviations

The following abbreviations are used in this manuscript:

LSTM    Long Short Term Memory
ANN     Artificial Neural Network
PYLB    Poyang Lake Basin
RMSE    root mean square error
NSE     Nash-Sutcliffe Efficiency

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
