# Peer review of "Comparison of Long Short Term Memory Networks and the Hydrological Model in Runoff Simulation"

_water, doi:10.3390/w12010175_

Round 1

Reviewer 1 Report

The article presents a discussion about the potential of LSTM in hydrology. The paper focus on an interesting issue and research results can be valuable. The paper is suitable topic for the readership of Water MDPI Journal. Therefore, I recommend a revision of this manuscript considering my comments below.

My comments are as follows:

- Line 31: Wouldn’t “model based” be “physically based model”?

- Line 46: “ML” has not been defined before;

- Figure 1: It is written “gauging” in the legend and inside Figure 1 (B);

- Section 3.6 (Hydrological Model): Although this SWAT model has already been used and calibrated elsewhere, this section should give more details of the hydrological modelling;

- Methodology: This section is mixing the methodology with the results. Where is the Results Section?

- Conclusions could be better developed.

Reviewer 2 Report

Please revise the paper based on the attached comments. Moderate revision is needed. 

Round 2

Reviewer 1 Report

The article presents a discussion about the potential of LSTM in hydrology and has been nicely revised. The paper focus on an interesting issue and research results can be valuable. The paper is suitable topic for the readership of Water MDPI Journal. Therefore, I recommend to accept this manuscript in present form.